

# Master regulator genes and their impact on major diseases

Wanwan Cai[1,*], Wanbang Zhou[2,*], Zhe Han[3], Junrong Lei[2], Jian Zhuang[4], Ping Zhu[4], Xiushan Wu[1] and Wuzhou Yuan[1]

[1] The Center for Heart Development, State Key Laboratory of Development Biology of Freshwater Fish, Key Laboratory of MOE for Development Biology and Protein Chemistry, College of Life Sciences, Hunan Normal University, Changsha, Hunan, China

[2] College of Physical Education, Hunan Normal University, Changsha, Hunan, China

[3] University of Maryland School of Medicine, Center for Precision Disease Modeling, Baltimore, MD, USA

[4] Guangdong Cardiovascular Institute, Guangdong Provincial People's Hospital, Guangdong Academy of Medical Sciences, Department of Cardiac Surgery, Guangzhou, Guangdong, China

[*] These authors contributed equally to this work.

## ABSTRACT

Master regulator genes (MRGs) have become a hot topic in recent decades. They not only affect the development of tissue and organ systems but also play a role in other signal pathways by regulating additional MRGs. Because a MRG can regulate the concurrent expression of several genes, its mutation often leads to major diseases. Moreover, the occurrence of many tumors and cardiovascular and nervous system diseases are closely related to MRG changes. With the development in omics technology, an increasing amount of investigations will be directed toward MRGs because their regulation involves all aspects of an organism's development. This review focuses on the definition and classification of MRGs as well as their influence on disease regulation.

## INTRODUCTION

Since the discovery of the master regulator genes (MRGs) and the powerful functions of these genes involved in all aspects of tissue and organ development, the study of MRGs have been more and more extensive, and an increasing number of new MRGs have been reported to play key roles in major clinical diseases. In the field of biomedicine, potential MRGs are generally analyzed based on the method of omic technologies, for instance, whole genome transcriptomics ChIPSeq and ATAC-Seq and well established bioinformatic analysis such as GSEA and its variants (*Alvarez et al., 2016*; *Boboila et al., 2018*; *Lefebvre et al., 2010*; *Tomljanovic et al., 2018*). Recent studies have pointed that the protein called myocyte enhancing factor 2C (MEF2C) is one of such master regulators involved in the pathogenesis of primary breast cancer. A systematic biological analysis of the transcriptional regulation activity of MEF2C and its target genes has revealed that this molecule induces collective responses leading to system-level gene expression deregulation and carcinogenesis (*Hernández-Lemus, Baca-López & Tovar, 2015*). A large number of clinical data from disease samples have been collected to calculate the potential MRGs

Corresponding author
Wuzhou Yuan, ywz@hunnu.edu.cn, yuanwuzhou@aliyun.com

in their pathological mechanisms. For example, in two breast cancer sample data sets, a systematic implementation of a series of algorithms is used to analyze the MRGs in potential primary breast cancer cells (*Baca-López et al., 2012*; *Lim, Lyashenko & Califano, 2009*; *Tapia-Carrillo et al., 2019*; *Tovar et al., 2015*). However, the definition of the MRG is still indistinct and imperfect, and a systematic and comprehensive review about MRGs is lacking. In this review, we proposed an updated definition and systematic classification of MRGs, and summarized the role of MRGs in major clinical diseases. The subject presented in this article is written in a descriptive manner instead of a systematic review so that clinicians outside our professional field can understand the basic characteristics of MRGs and their significant effects on clinical diseases.

## WHAT IS THE MASTER REGULATOR GENE?

The term "master regulator gene" introduced by Susumu Ohno in 1978, refers to "the gene at the top of the regulatory hierarchy, which should not be affected by the regulation of any other genes" (*Ohno, 1978*). However, with the increasingly extensive and in-depth study of master regulator genes (MRGs) in recent decades, this definition is no longer an absolute. Many studies have shown that some MRGs can be regulated by others. For example, *mdm2* is the master regulator of tumor suppressor protein p53 (*Momand, Wu & Dasgupta, 2000*), while the *p53* gene is a master regulator of diverse cellular processes and a potential therapeutic target for cancer (*Farnebo, Bykov & Wiman, 2010*); and *snai1* is the master regulator of epithelial-mesenchymal transition, but it is regulated by *Pak1* through phosphorylation (*Takahashi et al., 2013*), which implicates *Pak1* as a master regulator of epithelial-mesenchymal transition (*Yang et al., 2005*).

It has been reported that MRGs play a key role via multiple signal pathways. For example, adenosine monophosphate-activated protein kinase (AMPK) regulates the energy balance inside cells by inhibiting adenosine triphosphate (ATP) consumption in the anabolic pathway and enhancing ATP synthesis in the catabolic pathway. When activated by external metabolic pressure, AMPK regulates a complex downstream signal cascade, promoting efficient energy production within the cells (*Witczak, Sharoff & Goodyear, 2008*). Another example is the phosphoinositide 3-kinase (PI3K)/protein kinase B (AKT)/mammalian target of rapamycin (mTOR) signaling pathway. Although this pathway is considered as a master regulator for cancer (*Schaefer, Steiner & Lengerke, 2020*; *Xia & Xu, 2015*), mTOR is also considered as a MRG of metabolism (*Kim & Guan, 2015*; *Zeng, 2017*). Furthermore, it has been reported that the genes for the three transcription factors Sox2, Oct3/4, and Nanog have been identified as the MRGs that regulate mammalian embryogenesis, embryonic stem cell self-renewal, and pluripotency. These MRGs can bind to enhancer elements in pluripotent embryonic stem cells (ESCs) and recruit mediators to form unusual enhancer domains, which are called super-enhancers. When the MRGs and mediators are simultaneously occupied, the expression programs for most genes in ESCs become co-activated (*Rizzino, 2008*; *Whyte et al., 2013*). Phenotypic conditions in living cells are largely determined by the interplay of a multitude of genes and their protein products, which form a gene regulatory network (GRN), and MRGs are the key players in GRNs. Gene regulatory

network analysis have shown that different levels of gene regulation are not only related but strongly coupled (*Hernández-Lemus, Baca-López & Tovar, 2015*). To summarize, MRGs can be updated as genes or signaling pathways that are expressed at the inception of a developmental lineage or a specific cell type, participate in the specification of that lineage by regulating multiple downstream genes' expression either directly or via interacting with other master regulator genes or signaling pathways to form super-enhancers, and critically, when misexpressed, will lead to uncontrolled expression of downstream target genes and MRGs, and have the ability to respecify the fate of cells destined to form other lineages, causing more abnormal development of tissues and organs.

## SURVEY METHODOLOGY

A survey of >2,000 articles was carried out using the National Center for Biotechnology Information PubMed database (https://www.ncbi.nlm.nih.gov/pubmed/) by searching the keyword "master regulator gene". After screening the contents of the abstracts of these literatures, we found that more than 900 articles quoting MRGs covered most species. Key words were extracted and recorded during the abstract reading, including the properties of the MRGs, the signaling pathways involved, the tissues or organs involved, and the diseases caused, etc. All the data was collated and considered effective. If multiple references mentioned a same MRG, we selected recently published papers or well-known journals for reference. These MRGs were systematically classified as either (1) whole-family MRGs, (2) signal pathway MRGs, or (3) tissue- or organ-specific MRGs.

## OVERVIEW OF MRGS

Family MRGs refer to a gene family where all members are MRGs. There are two types: either all members have the same function, such as the HOX, MTA, and SREBP families; or different members in the same family may possess different functions, such as the GATA gene family. The HOX family MRGs are all involved in developmental processes, such as embryogenesis and hematopoiesis (*Candini et al., 2015*; *Grier et al., 2005*; *Magnusson et al., 2007*; *McGonigle, Lappin & Thompson, 2008*; *Rice & Licht, 2007*; *Vogel et al., 2016*; *Zhang et al., 2015*). In mammals, the HOX network consists of 39 genes that exhibit a high degree of sequence similarity, particularly in the homeobox domain. Homeobox genes function as master regulatory transcription factors during development, and their expression is often altered in cancer (*Brotto et al., 2020*; *Li et al., 2020*; *Qu et al., 2019*). Many of the chromosomal translocations associated with acute leukemias involve HOX genes, such as mixed lineage leukemia, which leads to the inappropriate expression of specific HOX gene subsets (*Collins & Thompson, 2018*; *Dickson, Lappin & Thompson, 2009*). In the GATA family, where each member has a different function, GATA1 and GATA2 regulate erythropoiesis and hematopoiesis as MRGs (*Bresnick & Johnson, 2019*; *Castaño et al., 2019*; *Gutiérrez et al., 2020*; *Kang et al., 2012*; *Katsumura et al., 2018*; *Katsumura et al., 2014*; *Leonards et al., 2020*; *Philipsen, 2013*; *Siegwart et al., 2020*), GATA3 is an immune response MRG (*El-Arabey et al., 2020*; *Li, Campos & Iida, 2015*; *Mirlekar, 2020*; *Nicol et al., 2016*; *Nomura et al., 2019*), and GATA4 regulates embryonic pancreas development

(*Kondratyeva et al., 2017*). Table 1 lists 18 major family MRGs. Among them, the CDX, CDK, HSF, MTA, SREBP, Rho, HNF, IL families and the Rab GTPase superfamily contain genes with the same functions. In the PLK, PAX, TBX, SOX, RUNX, IRF, BCL, and C/EBP families, each family member shares similar functions but also performs their own distinct role. In Fig. 1, we have summarized typical family MRGs involved in regulation at the cellular level, including CDK Family, Rho Family and PLK Family involved in cell cycle regulation, and BCL Family involved in cell apoptosis, etc. Figure 2 summarizes the Family MRGs involved in tissue and organ development, including PAX Family involved in eye development, TBX Family involved in heart development, etc.

The second type of MRGs is signaling pathways MRGs. In this type, either one of the members in the signal pathway is the MRG, such as AMPK from the AMPK signal pathway, which is known as a master regulator of cellular energy metabolism due to its role in regulating glucose, lipid, and protein metabolism. AMPK is an evolutionarily conserved master regulator of metabolism and a therapeutic target in type 2 diabetes. As an energy sensor, AMPK activity is responsive to both metabolic inputs, i.e., the ratio of AMP to ATP and numerous hormonal cues (*Cunningham et al., 2014*; *Witczak, Sharoff & Goodyear, 2008*). Or more commonly, members of the whole signaling pathway cooperate with each other as MRGs to regulate the development of a series of tissues and organs. For example, the mTOR signaling pathway is a master regulator of cell growth, proliferation and survival, metabolism, and skeletal muscle production in eukaryotes (*Donnelly et al., 2017*; *Zeng, 2017*). mTOR belongs to the PI3K-related protein kinase family. The mTOR signaling pathway plays a crucial role in the functional recovery of central nervous system trauma, especially for axon regeneration and autophagy, which has an extensive association with apoptosis. Significantly, this pathway is receiving novel concern for its role in the repair and regeneration of traumatic central nervous system injuries, such as traumatic brain injury and spinal cord injury (*Lin, Huo & Liu, 2017a*). The novel concern for mTOR is also because it is a master regulator of the inflammatory response in immune and non-immune cells and implicated in a number of chronic inflammatory diseases, especially rheumatic diseases, such as systemic lupus erythematosus, rheumatoid arthritis, systemic sclerosis, sjogren syndrome and seronegative spondyloarthropathy (*Suto & Karonitsch, 2020*). mTOR signaling pathway acts as a master regulator in memory $CD8^+$ $T^-$cells, Th17, and NK cells development and their functional properties (*Rostamzadeh et al., 2019*). Researchers used RNAi system to specifically knockdown mTOR, raptor, S6K1, eIF4E, and FKBP12 expressions in antigenmune $CD8^+T^-$cells and the results have demonstrated that mTOR acts as the key regulator of memory $CD8^+T^-$ cells differentiation. When mTOR or raptor is knocked down, the expression levels of memory $T^-$ cell markers CD127, CD62L, Bcl-2, and CD27 are remarkably elevated. Significant increases in memory $CD8^+T^-$ cells differentiation after knockdown of S6K1 and eIF4E showed that mTOR exerted its effect through these two downstream molecules (*Araki et al., 2009*).

The major signaling pathways MRGs are presented in Table 2. For example, the transforming growth factor (TGF) β signaling pathway is the master regulator of the respiratory system, epithelial-mesenchymal transition and metastasis, and cancer development; Hedgehog signaling is the master regulator of cell differentiation; and the

**Table 1** Summary of family MRGs and their related functions.

| Family MRGs | Members | Functions |
|---|---|---|
| CDX Family | all | master regulator of HOX family gene (*Frohling et al., 2007*; *Rawat, Humphries & Buske, 2012*; *Shiotani et al., 2008*) |
| CDK Family | CDK1, CDK2, Cdc5 | master regulator of cell cycle regulation (*Botchkarev & Haber, 2018*; *Hinds, 2003*; *Satyanarayana & Kaldis, 2009*) |
| HSFs Family | HSF1, HSF2, HSFA1 | master regulator of heat shock reaction (*Filone et al., 2014*; *Liu & Charng, 2012*; *Qiao et al., 2017*; *Shinkawa et al., 2011* |
| MTA Family | all | master regulator of the occurrence and metastasis of cancer (*Du et al., 2017*; *PA, 2014*; *Zhu et al., 2009*) |
| SREBP Family | all | master regulator of lipid homeostasis (*Gong et al., 2016*; *Krycer et al., 2010*; *Madison, 2016*) |
| Rho Family | Including RhoA, Rac1 and Cdc42 proteins, and so on | master regulator for a large number of cell functions, including control of cell morphology, cell migration and polarity, transcriptional activation and cell cycle progression (*Bai et al., 2015*; *Colomba & Ridley, 2014*; *Costa et al., 2011*; *PA, 2014*; *Singh et al., 2019*; *Watanabe, Takano & Endo, 2006*; *Zago et al., 2019*) |
| HNF (Hepatocyte nuclear factor) Family | IncludingHNF1A/B, HNF4alpha, HNF6 | master regulator of pancreas and liver differentiation (*Alder et al., 2014*; *Janky et al., 2016*; *Kondratyeva et al., 2017*; *Odom et al., 2004*; *Sandovici et al., 2013*) |
| IL(interleukin) Family | Including IL-1, IL-2, IL-6, IL-7, IL-10, IL-12, IL-21, IL-23, IL-27, ILC3, and so on | master regulator of inflammation or immunity (*Fry & Mackall, 2001*; *Langrish et al., 2004*; *Neurath, 2007*; *Qin et al., 2017*; *Rojas et al., 2017*; *Sharma, Fu & Ju, 2011*; *Waldner & Neurath, 2014*; *Wilson & Esposito, 2009*; *Zhou & Sonnenberg, 2020*) |
| Rab GTPases Superfamily | Including Rab5, Rab7b, Rab11 GTPase, and so on | master regulator of cell membrane transport (*Distefano et al., 2015*; *Ishida, E. Oguchi & Fukuda, 2016*; *Pfeffer, 2017*; *Qi et al., 2015*; *Wu et al., 2015*) |
| The MiTF/TFE Family of Transcription Factors | MITF, TFEB, TFE3, TFEC | Master Regulators of Organelle Signaling, Metabolism, and Stress Adaptation (*Slade & Pulinilkunnil, 2017*) |
| PLK Family | all | master regulator of cell division |
| | PLK1 | master regulator of mitotic related kinases (*Combes et al., 2017*) |
| | PLK4 | master regulator of the formation of centrioles (*Levine & Holland Andrew, 2014*; *Shaheen et al., 2014*) |
| PAX Family | all | master regulator of development and tissue homeostasis (*Relaix, 2015*) |
| | Pax5 | master regulator of b-cell development and leukemia (*Medvedovic et al., 2011*; *Nebral et al., 2009*) |
| | Pax6 | master regulator of ganglion cells of the retina and eye development (*Albert et al., 2013*; *Shubham & Mishra, 2012*) |
| TBX Family | TBX1 | master regulator of muscle differentiation (*Chen et al., 2009*) |
| | TBX5 | master regulator of heart development (*Boogerd & Evans, 2016*) |
| | TBX21 | master regulator of Th1 cell development (*Nicol et al., 2016*; *Stolarczyk, Lord & Howard, 2014*; *Wilkie et al., 2016*) |

**Table 1** (*continued*)

| Family MRGs | Members | Functions |
|---|---|---|
| | SOX2 | master regulator of mammalian embryogenesis, embryonic stem cell self-renewal and pluripotency (*Rizzino, 2008*; *Whyte et al., 2013*) |
| | SOX3 | master regulator of innate immunity (*Doostparast Torshizi & Wang, 2017*) |
| | SOX4 | master regulator of EMT (epithelial-mesenchymal transition) (*Lourenço & Coffer, 2017*; *Tiwari et al., 2013*) |
| | SOX5, SOX6 | the interaction with SOX9 is a master regulator of cartilage development (*Ma et al., 2016*, *Suzuki et al., 2012*; *Vivekanandan et al., 2015*) |
| SOX Family | | master regulator of testis differentiation pathway (*Jakob & Lovell-Badge, 2011*; *Mork & Capel, 2010*; *Kozhukhar, 2012*) |
| | SOX9 | master regulator of fibroblast differentiation (*Noizet et al., 2016*) |
| | | master regulator of pancreatic program (*Julian, McDonald & Stanford, 2017*; *Seymour, 2014*) |
| | SOXB1, SOXE, SOXF | master regulator of cell fate (*Julian, McDonald & Stanford, 2017*) |
| RUNX Family | RUNX1 | master regulator of adult hematopoiesis (*Ichikawa et al., 2004*; *Wehrspaun, Haerty & Ponting, 2015*; *Wu et al., 2014*) |
| | RUNX2 | master regulator of osteoblast lineage (*Liu et al., 2017*; *Wysokinski, Pawlowska & Blasiak, 2015*) |
| | IRF-1 | master regulator of cross talk between macrophage and L929 fibrosarcoma cells (*Nascimento et al., 2015*) |
| | IRF4 | master regulator of human periodontitis (*Sawle et al., 2016*) |
| IRF Family | IRF7 | master regulator of IFN-I, virus-induced cytokine (*Hu et al., 2011*; *Lu et al., 2015*; *Wang et al., 2013*) |
| | IRF8 | master regulator of monocytes and dendritic cells development (*Tamura, 2017*) |
| | BCL-2 | master regulator of apoptosis (*Chen et al., 2012*; *Häcker & Vaux, 1995*) |
| | BCL-6 | master regulator of Tfh cell differentiation (*Matsumoto et al., 2017*) |
| BCL Family | BCL11B | master regulator of T cell (Th) differentiation (*Inoue et al., 2016*) |
| | BCL–2–like 10 | master regulator of Aurora kinase a mouse oocytes (*Lee et al., 2016*) |
| | C/EBPα | master regulator of the bone marrow progenitor cells and fat formation (*Ding et al., 2011*; *Okuno, Inoue & Imai, 2013*) |
| C/EBP Family | C/EBPbeta | master regulator of physiological cardiac hypertrophy (*Molkentin Jeffery, 2011*) |
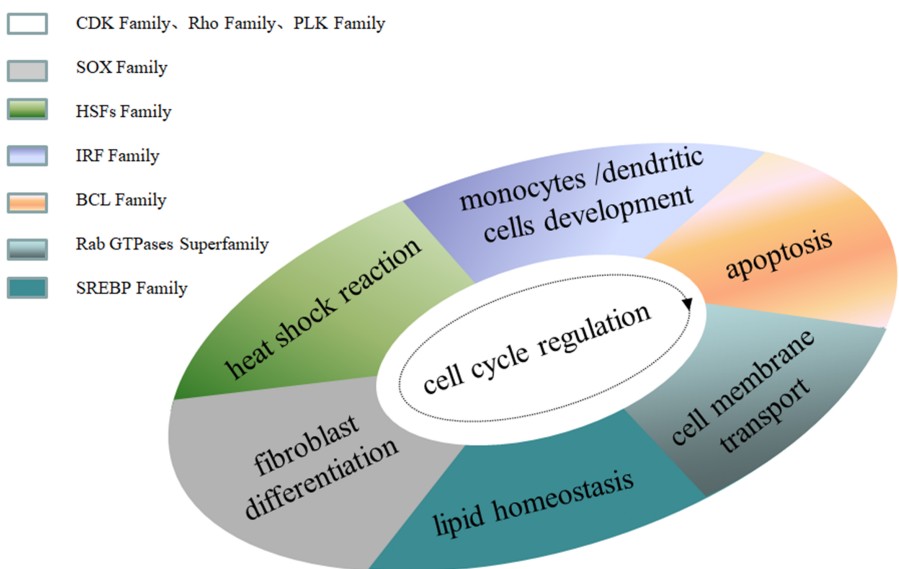

**Figure 1   Family MRGs associated with cellular level regulation. .**

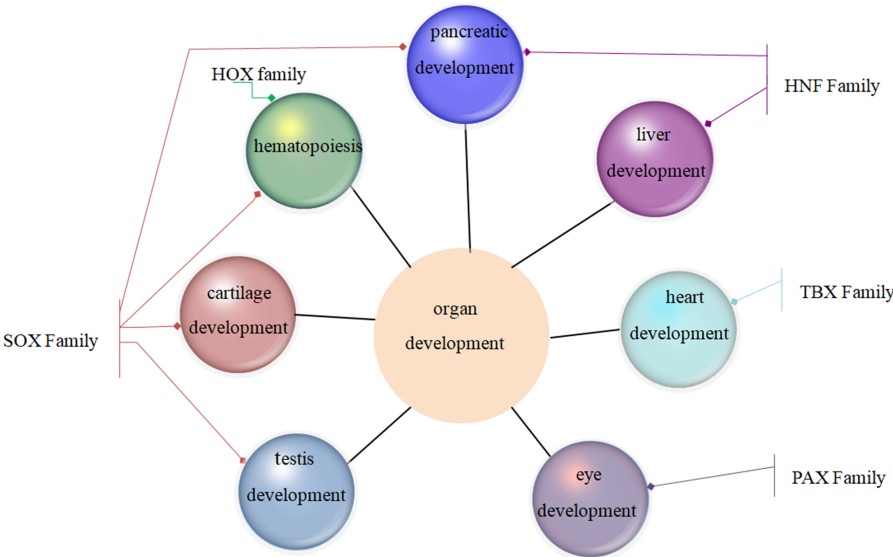

**Figure 2   Family MRGs involved in tissue and organ development. .**

NF-κB signaling pathway is the master regulator of innate immune and inflammatory signals. It is noteworthy that the Wnt signaling pathway is not only the master regulator of cell development, cell polarization, and brain invasion but also the master regulator of liver-region and multiple renin-angiotensin system genes.

The third type of MRGs is tissue- or organ-specific MRGs that regulate the development of different tissue and organ systems. Table 3 summarizes the MRGs associated with

**Table 2  Summary of the important signaling pathways MRGs.**

| Signaling pathway | Master regulator gene | Functions |
|---|---|---|
| TGF-β signaling pathway | TGF-β signaling pathway | master regulator of the respiratory system, epithelial-mesenchymal transition and metastasis, and cancer development, etc (*Fazilaty et al., 2013*; *Solomon et al., 2010*; *Zhou et al., 2014*) |
| PI3K-AKT-mTOR signaling pathway | PI3K-AKT-mTOR signaling pathway | master regulator of cancer (*Xia & Xu, 2015*) |
| Hedgehog (Hh) signaling pathway | Hedgehog (Hh) signaling pathway | master regulator of cell differentiation (*Peng & Joyner, 2015*) |
| NF-kappaB signaling pathway | NF-kappaB signaling pathway | master regulator of innate immunity and inflammatory signaling (*Krappmann et al., 2004*; *Matroule, Volanti & Piette, 2006*; *Schnappauf & Aksentijevich, 2020*; *Zeitz et al., 2017*) |
| | Wnt signal pathway | master regulator of cell development and cell polarization (*Gómez-Orte et al., 2013*) |
| Wnt signaling pathway | Wnt5a | master regulator of brain invasion (*Binda et al., 2017*) |
| | Wnt/β-catenin | master regulator of the liver region and multiple RAS (renin-angiotensin system) genes (*Torre, Perret & Colnot, 2010*) |
| | Notch | The fate of arteriovenous-lymphatic endothelial cells is regulated by the master regulator of Notch, COUP-TFII, and Prox1 (*Kang et al., 2010*) |
| Notch signaling pathway | Notch3 | master regulator of neuroblastoma movement (*Van Nes et al., 2013*) |
| Yap signaling pathway | Yap1 | master regulator of endometriosis (*Lin et al., 2017b*) |
| Hypoxia signaling pathway | HIF1, HIF-1α \ HIF-2α | master regulators of the adaptive response to hypoxia (*Lu & Kang, 2010*; *Schönenberger & Kovacs, 2015*; *Xiao, 2015*; *Zhao et al., 2020*) |

tissue/organ specificity, among which SCL/TAL1, VEGF, and PU.1 are the MRGs of hematopoiesis; Sim1 and Gcm are the MRGs of *Drosophila* neurodevelopment; FOXM1, Blimp1, Oct4, and Myc are the MRGs that regulate the cell cycle, B-cell differentiation to plasma cells, embryonic stem cells, and cell performance, respectively; CTCF is the MRG of human epigenetic and genomic spatial tissue; and FOXj1 is the MRG of the ciliary formation program. In bacteria, the MRGs include SinR, CtrA, FlhDC, Fur, CsgD, Spo0A, CcpA, LuxR, and WOR1. Details and other tissue- and organ-specific MRGs are listed in Table 3.

# REGULATION OF MAJOR DISEASES BY THE MRGS

Since MRGs can concurrently regulate the expression of hundreds of genes, their expression levels must be tightly controlled, otherwise, misexpression or overexpression will exert a considerable impact on the development of affected organisms, resulting in runaway or uncontrolled metabolism and abnormal development in humans.

## MRGs regulation of tumors

MRGs have been implicated in the occurrence of different tumors, including gum germ cell tumors, ovarian cancer, colon cancer, rectal cancer, and lung cancer. For example,
**Table 3  Summary of reported MRGs and their related functions.**

| MRGs | Related functions |
|---|---|
| SCL/TAL1 | master regulator of the adult hematopoietic (*Courtial et al., 2012*; *Wehrspaun, Haerty & Ponting, 2015*) |
| VEGF | master regulator of mucosal immunity driving angiogenesis (*Danese, 2008b*) |
| PU.1 | master regulator of hematopoiesis and bone marrow (*Yang et al., 2012*) |
| Sim1 | master regulator of *Drosophila* neurogenesis (*Eaton & Glasgow, 2006*) |
| Gcm | master regulator of nervous system development in *Drosophila*, parathyroid development, master regulator of expression and function regulation in mammals (*Cattenoz & Giangrande, 2016*) |
| FOXM1 | master regulator of different stages of the cell cycle (*Jeffery et al., 2017*; *Zona et al., 2014*) |
| Blimp1 | master regulator of B cell differentiation into plasma cells (*John & Garrett-Sinha, 2009*; *Vrzalikova, Woodman & Murray, 2012*) |
| Oct4 | master regulator of embryonic stem cell self-renewal and pluripotency (*Samardzija et al., 2017a*) |
| Myc | master regulator of cell performance (growth, proliferation, stem cell pluripotency, ribosomal biogenesis, etc.) (*Grifoni & Bellosta, 2015*; *Holmberg Olausson, Nistér & Lindström, 2012*; *Kazan & Manners, 2013*) |
| HIF | master regulator of cellular responses to hypoxia (*Liu, Semenza & Zhang, 2015*; *Semenza, 2014*; *Semenza, 2017*) |
| CTCF | master regulator of human epigenetics and genomic spatial organization (*Golan-Mashiach et al., 2012*) |
| FOXj1 | master regulator of cilia generation program (*Yu et al., 2008*) |
| SinR | master regulator of Bacillus subtilis biofilm formation (*Chu et al., 2006*; *Stowe et al., 2014*) |
| CtrA | master regulator of the cell cycle of the bacillus (*Gora et al., 2010*; *Laub et al., 2002*; *Pini et al., 2015*) |
| FlhDC | master regulator of flagellar genes (*Chatterjee, Cui & Chatterjee, 2015*; *Cui et al., 2008*; *Stafford, Ogi & Hughes, 2005*) |
| Fur | master regulator of iron metabolism in Gram-negative bacteria (*González et al., 2012*; *Huja et al., 2014*) |
| CsgD | master regulator of *E. coli* biofilm formation (*Ogasawara, Yamamoto & Ishihama, 2010*; *Wen et al., 2017*) |
| Spo0A | master regulator of the pathogenesis of *Bacillus subtilis* spore formation (*Fujita & Losick, 2005*; *Wolański & Jakimowicz, 2014*) |

**Table 3** (*continued*)

| MRGs | Related functions |
| --- | --- |
| CcpA | master regulator of carbon catabolism regulation in *Bacillus* (*Muscariello et al., 2013*; *Weeks et al., 2012a*; *Weeks et al., 2012b*) |
| LuxR | master regulator of quorum sensing (*Ball, Chaparian & van Kessel, 2017*; *Pompeani et al., 2008*) |
| WOR1 | master regulator of white and opaque phenotypes of Candida albicans (*Zhang et al., 2014*) |
| P53 | master regulator of human malignant tumors (*Farnebo, Bykov & Wiman, 2010*; *Resnick et al., 2005*) |
| P63 | master regulator of epidermal development and differentiation (*Soares & Zhou, 2018*) |
| Nrf2 | master regulator of redox homeostasis (*Basak et al., 2017*; *Cores et al., 2020*; *Hayes & Dinkova-Kostova, 2017*) |
| MITF | master regulator of melanocyte development (*Levy, Khaled & Fisher, 2006*) |
| TFEB | master regulator of lysosomal biogenesis and autophagy (*Medina et al., 2015*; *Settembre et al., 2011*) |
| MyoD | master regulator of skeletal muscle gene expression programs (*Aziz, Liu & Dilworth, 2010*; *Sunadome et al., 2014*) |
| MicroRNAs (miR-10b*,miR21, miR-31,miR153, miR156, etc.) | master regulator of gene expression in many physiological and pathological processes (*Biagioni et al., 2012*; *Datta & Paul, 2015*; *Kaul & Krams, 2015*; *Liang et al., 2020*; *Miranda et al., 2010 Schmittgen, 2010*; *Stief et al., 2014*; *Voorhoeve, 2010*) |
| PGC-1 $\alpha$ | master regulator of mitochondrial gene expression (*Fernandez-Marcos & Auwerx, 2011*; *Zhu et al., 2009*) |
| Prox1 | master regulator of lymphatic endothelial cell differentiation (*Hong & Detmar, 2003*; *Kang et al., 2010*; *Ke & Yang, 2017*) |
| AphA | master regulator of quorum sensing (*Sun et al., 2012*; *Van Kessel et al., 2013*) |
| PPARgamma | master regulator of fat formation (*Lehrke & Lazar, 2005a*; *Sunadome et al., 2014*) |
| foxp3 | master regulator of regulatory T (Treg)cell development and function (*Liston, 2010*; *Thornton & Shevach, 2019*) |
| ComK | master regulator of late competence genes (*Jaskólska & Gerdes, 2015*; *Ogura, Hashimoto & Tanaka, 2002*) |

SOX9, GATA4, PDX1, PTF1a, HNF1b, and GRP78 are master regulators of pancreatic cancer (*Kondratyeva et al., 2017*); while Srebp2 (*Krycer et al., 2010*) and E2F8 (*Rohde et al., 1996*) are MRGs of prostate cancer; and CDX2 is the master regulator of gastric cancer (*Shiotani et al., 2008*). Nuclear receptors are liver cancer-related (*Jakobsson et al., 2012*); PD-L1, TGF-β1, and IL-10 are the master regulators of cervical cancer (*Qin et al., 2017*); and Oct4A is the master regulator of ovarian cancer (*Samardzija et al., 2017*). Analysis of master regulatory genes may help to understand the most upstream events in phenotypic development, particularly those related to cancer biology.

The most extensively studied MRGs are associated with breast cancer and leukemia. Breast cancer is the most common malignant tumor in women. It has been reported that RUNX1 encodes the transcription factor of the RUNX family, a new mutation in RUNX gene was discovered in human breast cancer. It was reported that RUNX1 was expressed in all subpopulations of mouse mammary epithelial cells (MECs) except for secretory alveolar cells. The conditional knockout of RUNX1 in the MECs resulted in the reduction of luminal MECs. Mainly due to a significant reduction in estrogen receptors (ERs), this phenotype could be rescued by the absence of Trp53 or Rb1. The underlying molecular mechanism was explained by RUNX1 inhibiting the expression of *Elf5* (the dominant gene in alveolar cells) and regulating the involvement of mature transcription factor or cofactor genes (such as *Foxa1* and *Cited1*) in the processes of ER synthesis (*Van Bragt et al., 2014*). Many other MRGs have been reported to be associated with the development of breast cancer, including the HOX gene family, SOX4, RUNX2, AMPK, p53, TGF-β, microRNA, KDM4B, p16INK4A, BACH1, Snai1, HMGA1, SATB1, HSP90, TRB3, Ddx5 and Ddx17, FGFR2, and AGTR2 (Table S1).

Another type of widely studied cancer is leukemia, a malignant clonal disease of hematopoietic stem cells. Due to uncontrolled proliferation, differentiation disorder, and blocked apoptosis, clonal leukemia cells proliferate and accumulate in the bone marrow and other hematopoietic tissues, infiltrate other non-hematopoietic tissues and organs, and inhibit normal hematopoietic function. Acute lymphoblastic leukemia (ALL) is the most common form of childhood cancer and is characterized by impaired lymphocyte differentiation, resulting in the accumulation of immature progenitor cells in the bone marrow, peripheral blood, and occasionally the central nervous system. Although ALL cure rates are close to 90%, it remains the leading cause of cancer-related mortality in children and young adults. Another extremely prevalent form of leukemia is B-cell precursor (BCP)-ALL, which represents 85% of cases, while the remaining 15% involve T-cell precursors. It was reported that BCP-ALL might be caused by the synergistic regulation of transcription factors, such as RUNX1, IKZF1, E2A, EBF1, and PAX5 (*Tijchon et al., 2012*). The other MRGs associated with leukemia include HOX, GATA, CDX, Pax, C/EBPistic genetic lesions, and key transcriptional targets and pathways (Table S1).

## Influence of MRGs on cardiovascular diseases

Because cardiovascular disease is the leading cause of death in humans, elucidation of the associated role of MRGs is of immense clinical and social value for the effective prevention and treatment of cardiovascular diseases. The MRGs related to heart disease (Table 4)
**Table 4  Summary of MRGs related to heart disease.**

| MRGs | Cardiovascular disease type |
|---|---|
| TBX5, NuRD | Congenital heart disease (*Boogerd & Evans, 2016*) |
| SREBP | Treatment of cardiac metabolic diseases (*Krycer et al., 2010*) |
| VEGF | Vascular disease (*Danese, 2008b*; *Gianni-Barrera et al., 2014*) |
| MyoD | Heart disease (*Kojima & Ieda, 2017*) |
| PPAR $\gamma$ | Obesity, diabetes and cardiovascular disease (*Lee & Ge, 2014*; *Lehrke & Lazar, 2005*) |
| PKCδ | Thrombosis complications (*Fischer, 2009*) |
| SCL/TAL1 | Anemia patient (*Fujiwara, 2017*) |
| Class IB phosphoinositide 3-kinase p110s | Heart disease (*Perino, Ghigo & Hirsch, 2010*) |
| PI3K | Heart failure (*Weeks et al., 2012a*) |
| SOX9 and myocardin | Atherosclerosis, vascular calcification (*Xu et al., 2012*) |
| Klotho | Cardiovascular diseases (*Moe Sharon, 2012*) |
| PITX2 | Atrial fibrillation (AF) is the most common persistent Arrhythmia (*Li, Dobrev & Wehrens, 2016*) |
| FLYWCH1, PSORSIC3, G3BP1 | Coronary artery disease (CAD) (*Foroughi Asl et al., 2015* |
| Thyroid hormones (THs) | Cardiovascular diseases (*Rajagopalan & Gerdes, 2015*) |
| CST | Cardiovascular diseases (CVD) (*Sushil, Malapaka & Nitish, 2018*) |
| Etv2 | Chronic vascular disease (*Garry, 2016*) |

include TBX5, NuRD, SREBP, MyoD, Class IB PI3K p110 genetic lesions, PI3K, and PITX2, which mainly regulate congenital heart disease, metabolic heart disease, heart failure, arrhythmia, etc. Vascular-related MRGs, which include PKCδ, VEGF, SCL/TAL1, PPAR gamma, PGC-1alpha, SOX9, myocardin, FLYWCH1, PSORSIC3, G3BP1, and Etv2, mainly regulate thrombosis, anemia, atherosclerosis, vascular calcification, coronary artery disease, chronic vascular disease, etc. Others, like, Klotho, thyroid hormones and thyroid-stimulating hormone, and CST were also reported as master regulators of cardiovascular disease.

## Influence of MRGs on Nervous system diseases

Nervous system diseases refer to the diseases that occur in the central nervous system, peripheral nervous system and vegetative nervous system, with sensory, motor, consciousness and vegetative nervous dysfunction as the main manifestations, among which the central nervous system diseases are the most widely studied. The central nervous system disease generally refers to the central nervous system degenerative disease, which refers to a group of diseases produced by the chronic progressive degeneration of the central nervous system. Pathologically, there are neuronal degeneration and neuron loss in the brain and/or spinal cord. Major diseases include Parkinson's disease, the overall ischemia, stroke, epilepsy, Alzheimer's disease and Huntington's disease, etc. At present, many articles have clarified the important role of master regulator genes in neurodegenerative diseases. For example, REST, a major transcriptional regulator of neurodegenerative diseases, is

a transcriptional suppressor that silences target genes through epigenetic remodeling. REST and REST-dependent epigenetic remodeling provide a central mechanism critical to the progressive neuronal degeneration associated with neurologic disorders and diseases including global ischemia, stroke, epilepsy, Alzheimer's and Huntington's disease (*Hwang & Zukin, 2018*). NRF2 regulation processes as a source of potential drug targets against neurodegenerative diseases (*Buendia et al., 2016*; *Cores et al., 2020*). ZCCHC17 is a master regulator of synaptic gene expression in Alzheimer's disease (*Tomljanovic et al., 2018*). ATF2 and PARK2 are transcription factors that act as MRGs in Alzheimer's disease (*Vargas et al., 2018*). The ubiquitin-proteasome system is a master regulator of neural development and the maintenance of brain structure and function (*Luza et al., 2020*), etc. At present, it has not been reported that there is a specific drug effective for various neurological diseases in the world. For many patients, relevant drugs just only relieve symptoms rather than cure diseases, causing indelible damage to patients' physical and mental health. Exploring novel MRGs working on the nervous system and disclosing the molecular mechanism of nervous system diseases, may become the exciting expect to develop target drugs and therapeutic schedule to achieve special purpose for the treatment of patients.

There are still many references on the research of master regulatory genes and other human various diseases. For example, there are some reports on the progress of investigating the influence of MRGs on diseases such as inflammatory bowel disease (*Danese, 2008a*), cartilage disease (*Ma et al., 2016*), and human diseases related to fibroblasts (*Shenoy et al., 2014*). Thus, the influence of MRGs on human diseases has permeated every aspect, and MRGs play a vital role in the clinical research and treatment of human diseases. However, how the MRGs can be used more comprehensively to solve the therapy problems in human diseases is an arduous task at present.

## OUTLOOK

With the sustained development in omics technologies, research pertaining to MRGs will continue getting more concern and progress because the involvement of MRGs in all aspects of an organism's development is becoming apparent. Here we demonstrated that MRGs fell within three operating motifs: (1) whole-family MRGs, (2) signaling pathway MRGs, and (3) tissue- or organ-specific MRGs and updated the definition of MRGs as genes or signaling pathways that are expressed at the inception of a developmental lineage or a specific cell type, participates in the specification of that lineage by regulating multiple downstream genes' expression either directly or via interacting with other master regulator genes or signaling pathways to form super-enhancers, and critically, when misexpressed, will lead to uncontrolled expression of downstream target genes and MRGs, and have the ability to respecify the fate of cells destined to form other lineages, causing more abnormal development of tissues and organs. The formidable function of an MRG lies not only in its regulation of the concurrent expression of hundreds of genes but also the diversity of its functions on human diseases.

MRGs play important roles in the occurrence of various human diseases (such as cancer, cardiovascular diseases and neurological diseases) and exhibit a great potential to

be targets of gene therapies and drugs. Therefore, exploring the MRGs corresponding to the pathological mechanisms of different diseases is particularly critical. At present, there have been many reports on the analysis of potential MRGs through different calculation methods, and subsequent experimental verification, which greatly improves the process of discovering and determining MRGs in the pathogenesis. Of course, the use of MRGs for gene therapy or targeted drugs is still a huge challenge, and its clinical application is also a long process, which requires unremitting efforts of the medical research team. We believe that the day of technological breakthroughs of MRGs will definitely come.

### Funding

This study was supported in part by grants from the National Natural Science Foundation of China (Nos.: 81370451, 81470449, 81670290, 81570279), the Cooperative Innovation Center of Engineering and New Products for Developmental Biology of Hunan Province (No. 2013-448-6). The funders had no role in study design, data collection and analysis, decision to publish, or preparation of the manuscript.

### Grant Disclosures

The following grant information was disclosed by the authors:
National Natural Science Foundation of China: 81370451, 81470449, 81670290, 81570279.
Cooperative Innovation Center of Engineering.
New Products for Developmental Biology of Hunan Province: 2013-448-6.

### Competing Interests

The authors declare there are no competing interests.

### Author Contributions

- Wanwan Cai, Wanbang Zhou, Xiushan Wu and Wuzhou Yuan conceived and designed the experiments, performed the experiments, analyzed the data, prepared figures and/or tables, authored or reviewed drafts of the paper, and approved the final draft.
- Zhe Han performed the experiments, prepared figures and/or tables, authored or reviewed drafts of the paper, and approved the final draft.
- Junrong Lei, Jian Zhuang and Ping Zhu performed the experiments, prepared figures and/or tables, and approved the final draft.

### Data Availability

The raw data are available in the Table S1.

### Supplemental Information

Supplemental information for this article can be found online at http://dx.doi.org/10.7717/peerj.9952#supplemental-information.

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
