# Peer review of "Master regulator genes and their impact on major diseases"

_PeerJ, doi:10.7717/peerj.9952_

## Round 0.1 · original submission · Minor Revisions

Please address all the concerns raised by reviewers.

·

Basic reporting

The manuscript "Master regulator genes and their impact on major diseases" presents a revision of the literature on master regulator genes and its relationship to human diseases. The authors declared that the definition of MRG remains "ambiguous". Actually this is not the case.

Though it may seem strange, given the authors broad coverage of the literature, there are some key references missing. They refer to well known works that define MRGs in quantitative and measurable terms and applied them in biomedical research. These methods are based on the results of omic technologies, for instance whole genome transcriptomics, ChIPSeq and ATAC-Seq and well established bioinformatic analyses such as GSEA and its variants. We strongly encourage the authors to look up at the following references for analytical and computational methods and applications:

Lefebvre, C., Rajbhandari, P., Alvarez, M. J., Bandaru, P., Lim, W. K., Sato, M., ... & Basso, K. (2010). A human B‐cell interactome identifies MYB and FOXM1 as master regulators of proliferation in germinal centers. Molecular systems biology, 6(1), 377.

Lim, W. K., Lyashenko, E., & Califano, A. (2009). Master regulators used as breast cancer metastasis classifier. In Biocomputing 2009 (pp. 504-515).

Boboila, S., Lopez, G., Yu, J., Banerjee, D., Kadenhe-Chiweshe, A., Connolly, E. P., ... & Yamashiro, D. J. (2018). Transcription factor activating protein 4 is synthetically lethal and a master regulator of MYCN-amplified neuroblastoma. Oncogene, 37(40), 5451-5465.

Tomljanovic, Z., Patel, M., Shin, W., Califano, A., & Teich, A. F. (2018). ZCCHC17 is a master regulator of synaptic gene expression in Alzheimer’s disease. Bioinformatics, 34(3), 367-371.

Dinkelspiel, H., Iyer, A., Lefebvre, C., Wright, J., Lewin, S., Herzog, T., ... & Califano, A. (2013). Identification of master regulators of cisplatin resistance in ovarian cancer. Gynecologic Oncology, 130(1), e4-e5.

Lefebvre, C., Rajbhandari, P., Alvarez, M. J., Bandaru, P., Lim, W. K., Sato, M., ... & Basso, K. (2010). A human B‐cell interactome identifies MYB and FOXM1 as master regulators of proliferation in germinal centers. Molecular systems biology, 6(1), 377.

Tovar, H., García-Herrera, R., Espinal-Enríquez, J., & Hernández-Lemus, E. (2015). Transcriptional master regulator analysis in breast cancer genetic networks. Computational biology and chemistry, 59, 67-77.

Tapia-Carrillo, D., Tovar, H., Velázquez-Caldelas, T., & Hernandez-Lemus, E. (2019). Master regulators of signaling pathways: An application to the analysis of gene regulation in breast cancer. Frontiers in genetics, 10, 1180.

Hernández-Lemus, E., Baca-López, K., & Tovar, H. (2015). What makes a transcriptional master regulator? A systems biology approach. In Physical Biology of Proteins and Peptides (pp. 161-174). Springer, Cham.

Alvarez, M. J., Giorgi, F., & Califano, A. (2014). Using viper, a package for Virtual Inference of Protein-activity by Enriched Regulon analysis. Bioconductor, 1-14.

Baca-López, K., Mayorga, M., Hidalgo-Miranda, A., Gutiérrez-Nájera, N., & Hernández-Lemus, E. (2012). The role of master regulators in the metabolic/transcriptional coupling in breast carcinomas. PLoS One, 7(8), e42678.

Aside from considering these additional information sources, I would like to suggest the authors to include one or perhaps two figures as aids to explain in a semi-mechanistic way the functional roles of MRGs.

Aside from these conceptual and presentation issues, the paper may become a relevant source of information on the role of MRGs in biomedical research.

Experimental design

This is a "traditional" review paper (i.e. not a systematic revision nor a meta-analysis) so the "design" is quite straightforward. I only suggest the authors to revise some additional references and try to add some illustrative figures to broaden the scope and readability of their manuscript.

Validity of the findings

This is a review paper

Additional comments

Aside from a couple of comments already made, this looks like a timely contribution to a (slowly) nascent field.

Reviewer 2 ·

Basic reporting

The review talks about the Master Regulator Genes (MRG) in view of different disease conditions. The review is within the scope of Peer J. It describes many aspects of MRGS and I commend the authors for doing a large sum of literature review, although the English writing is not up to the mark of a reputed journal. Further, it fails to put forward any new insights or describe details in a critical manner. The content is mere description or paraphrasing of existing articles. The introduction section is small with very less background.

Experimental design

The authors did a good job in incorporating a large volume of references. It represents unbiased coverage of the subject. The writing is non-coherent in many instances. I mentioned few of those errors in general comment. I would encourage the authors to consult a science writer to improve the English. I would also encourage to use pictorial representations whenever possible to make it more interesting to the reader.

Validity of the findings

The review has a very short conclusion/outlook section that fails to identify any unresolved questions or gaps. The conclusion is an important part and the authors need to give more importance in writing this section. The writing in this section need major restructuring. For e.g.,line 219 “ ….gene regulation in all aspects of individual development is becoming apparent .“ – what does ‘individual development’ mean here ? Does the author mean development of an organism?
Also, the last sentence “ For the application of MRGs in clinical medical treatment, it is necessary to strengthen the in-depth understanding of disease-related MRGs to develop target-specific and more effective treatment programs.” This sentence does not convey the message well. I would not recommend starting the sentence with ‘For’. There are many such examples throughout the article.

Additional comments

There are some grammatical mistakes and other issues as described below:
Line 45 “ MRGs has been reported…….” : It should be MRGs have been.
Line 46 “ in major clinical diseases..” : Please remove one fullstop.
In section 3 Overview of MRGs: Inclusion of a figure showing the different MRG families in pictorial representation will make this section more interesting.
Line 138-153: The paragraph is filled with gene names and references. This make the section hard to read, I would suggest to use tables for such descriptions.
In Section 4, regulation of major diseases by MRGS is discussed but it is not clarified why the author choose to focus on only tumor and cardiovascular diseases while the MRGS can regulate many other diseases.
Line 171-172 “ RUNX family, which was recently discovered to be a new mutant gene in human…….” : new mutant gene is not a proper terminology, I would suggest writing - A new mutation in RUNX gene was discovered. Use of proper terms and structure of sentence is very important and the author should review this criteria throughout the article.
Line 179-180 “ MRGs have been reported as associated……”: MRGs have been reported to be associated.
Line 209-210 “ There are some reports on the……” : Please add references.
Line 209-214 : This paragraph closes the section ‘Regulation of Major Diseases by the MRGs” but the writing seems abrupt without continuation to the previous writing. Need rewriting.

In table 2: what does ‘whole signal pathway’ mean? I would suggest some alternative terminology.

·

Basic reporting

There is a growing interest in the concept of master regulator genes (MRGs). This review appears to be relevant in the current situation and for future considerations during efficacious treatment development for various clinical diseases. Introduction highlights the importance of MRGs, the lacuna in the area and the portion of coverage provided by this review to fulfil the deficit.
The review is well within the scope and it will be of interest to the readers of the journal. Although, there are many reviews covering a specific MRGs in diverse area, the current review is one of its kind that spans different aspects and provides classification of MRGs. However, the most recent literature available in diverse areas are missing. The specific details are mentioned later in the comments. It will be in the interest of authors to cite most recent literature to enhance the value of manuscript.

The review is well structured, and English is clear, unambiguous, and professional. There are some minor punctuation issues that needs to be corrected. The details of which are provided along with other comments.

Experimental design

The literature survey is unbiased and well organized into sections and sub-sections. However, to be comprehensive and up to date, recent developments in the area should be included. Authors have covered MRGs involvement in tumors and cardiovascular diseases. Recently, the MRGs in various diseases have been identified, which can be adequately covered.

Validity of the findings

The authors have systematically categorized MRGs and elaborated their involvement in various processes. MRGs involvement in tumors and cardiovascular diseases is well covered and the requirement of in-depth understanding for their application in clinical medical treatment is highlighted. To increase the impact of the manuscript, the MRGs involvement in other diseases can be incorporated. For instance, the information for MRGs in metabolic and neurodegenerative diseases can be covered.

Below are some recent advances in the area:
NRF2: Cores Á, Piquero M, Villacampa M, León R, Menéndez JC. NRF2 Regulation Processes as a Source of Potential Drug Targets against Neurodegenerative Diseases. Biomolecules. 2020 Jun 14;10(6):E904. doi: 10.3390/biom10060904. PMID: 32545924.
Nrf2-ARE pathway: Buendia I, Michalska P, Navarro E, Gameiro I, Egea J, León R. Nrf2-ARE pathway: An emerging target against oxidative stress and neuroinflammation in neurodegenerative diseases. Pharmacol Ther. 2016 Jan;157:84-104. doi: 10.1016/j.pharmthera.2015.11.003. Epub 2015 Nov 23. PMID: 26617217.
The Keap1-Nrf2 pathway: Deshmukh P, Unni S, Krishnappa G, Padmanabhan B. The Keap1-Nrf2 pathway: promising therapeutic target to counteract ROS-mediated damage in cancers and neurodegenerative diseases. Biophys Rev. 2017 Feb;9(1):41-56. doi: 10.1007/s12551-016-0244-4. Epub 2016 Dec 6. PMID: 28510041; PMCID: PMC5425799.
The ubiquitin proteasome system and schizophrenia: Luza S, Opazo CM, Bousman CA, Pantelis C, Bush AI, Everall IP. The ubiquitin proteasome system and schizophrenia. Lancet Psychiatry. 2020 Jun;7(6):528-537. doi: 10.1016/S2215-0366(19)30520-6. Epub 2020 Feb 12. PMID: 32061320.

Additional comments

General Comment: Overall, the study is interesting and a useful contribution to the area. The strength of the manuscript is the diverse coverage and systemic classification of the master regulator genes. A general comment on the manuscript is the need of citation of recent references. These changes will present the current scenario to the reader. However, there are some minor concerns, which are detailed below. A revision is recommended for improving the manuscript.

Page 3, Line 78-84: The authors have provided the definition that partly overlaps with already existing definition in an article (Chan SS, Kyba M. What is a Master Regulator? J Stem Cell Res Ther. 2013 May 4;3:114. doi: 10.4172/2157-7633.1000e114. PMID: 23885309; PMCID: PMC3718559.). The definition presented in the published article states that, “We propose that the best interpretation of the term master regulator is a gene that is expressed at the inception of a developmental lineage or cell type, participates in the specification of that lineage by regulating multiple downstream genes either directly or through a cascade of gene expression changes, and critically, when misexpressed, has the ability to respecify the fate of cells destined to form other lineages”.
Although in the next sentence, the authors have cited the above reference but only for a portion of its definition. It will be appropriate to provide due credits to what already exists and defined. Therefore, rephrasing is recommended. The addition to this existing definition by authors can be highlighted.

Page 3, Line 71: It will be appropriate to add recent reference for this pathway as a master regulator for cancer. "Schaefer T, et al. SOX2 and p53 Expression Control Converges in PI3K/AKT Signaling with Versatile Implications for Stemness and Cancer. Int J Mol Sci. 2020 Jul 11;21(14):E4902. doi: 10.3390/ijms21144902. PMID: 32664542."

Page 4, Line 99-105: Incorporation of the recent articles on HOX family and regulation mediated by them will be more relevant. For instance, the following articles are already available.
Brotto DB, et al. Contributions of HOX genes to cancer hallmarks: Enrichment pathway analysis and review. Tumour Biol. 2020;42(5):1010428320918050. doi:10.1177/1010428320918050
Collins EM, Thompson A. HOX genes in normal, engineered and malignant hematopoiesis. Int J Dev Biol. 2018;62(11-12):847-856. doi: 10.1387/ijdb.180206at. PMID: 30604854.
Li M, et al. Epigenetic upregulation of HOXC10 in non-small lung cancer cells. Aging (Albany NY). 2020 Jul 19;12. doi: 10.18632/aging.103597. Epub ahead of print. PMID: 32687064.
Qu X, et al. HOX transcript antisense RNA (HOTAIR) in cancer. Cancer Lett. 2019;454:90-97. doi:10.1016/j.canlet.2019.04.016

Page 4, Line 106-109: Similarly, in other sections please include recent articles for GATA family members. For, GATA1 and GATA 2:
Gutiérrez L, et al. Regulation of GATA1 levels in erythropoiesis. IUBMB Life. 2020;72(1):89-105. doi:10.1002/iub.2192
Leonards K, et al. Nuclear interacting SET domain protein 1 inactivation impairs GATA1-regulated erythroid differentiation and causes erythroleukemia. Nat Commun. 2020 Jun 12;11(1):2807. doi: 10.1038/s41467-020-16179-8. PMID: 32533074; PMCID: PMC7293310.
Siegwart LC, et al. The transcription factor NFE2 enhances expression of the hematopoietic master regulators SCL/TAL1 and GATA2. Exp Hematol. 2020 Jun 25:S0301-472X(20)30245-9. doi: 10.1016/j.exphem.2020.06.004. Epub ahead of print. PMID: 32593672.
Castaño J, et al. GATA2 Promotes Hematopoietic Development and Represses Cardiac Differentiation of Human Mesoderm. Stem Cell Reports. 2019 Sep 10;13(3):515-529. doi: 10.1016/j.stemcr.2019.07.009. Epub 2019 Aug 8. PMID: 31402335; PMCID: PMC6742600.
Bresnick EH, Johnson KD. Blood disease-causing and -suppressing transcriptional enhancers: general principles and GATA2 mechanisms. Blood Adv. 2019 Jul 9;3(13):2045-2056. doi: 10.1182/bloodadvances.2019000378. PMID: 31289032; PMCID: PMC6616255.
Katsumura KR, et al. Human leukemia mutations corrupt but do not abrogate GATA-2 function. Proc Natl Acad Sci U S A. 2018 Oct 23;115(43):E10109-E10118. doi: 10.1073/pnas.1813015115. Epub 2018 Oct 9. PMID: 30301799; PMCID: PMC6205465.

For GATA 3:
Nomura S, et al. Pyrrothiogatain acts as an inhibitor of GATA family proteins and inhibits Th2 cell differentiation in vitro. Sci Rep. 2019 Nov 22;9(1):17335. doi: 10.1038/s41598-019-53856-1. PMID: 31758034; PMCID: PMC6874683.
El-Arabey AA, et al. GATA3 as a master regulator for interactions of tumor-associated macrophages with high-grade serous ovarian carcinoma. Cell Signal. 2020 Apr;68:109539. doi: 10.1016/j.cellsig.2020.109539. Epub 2020 Jan 11. PMID: 31935430.
Mirlekar B. Co-expression of master transcription factors determines CD4+ T cell plasticity and functions in auto-inflammatory diseases. Immunol Lett. 2020 Jun;222:58-66. doi: 10.1016/j.imlet.2020.03.007. Epub 2020 Mar 24. PMID: 32220615.

Page 5, Line 126-128: The novel concern for mTOR is also because it is a master regulator of the inflammatory response in immune and non-immune cells and implicated in a number of chronic inflammatory diseases, especially rheumatic diseases, such as systemic lupus erythematosus, rheumatoid arthritis, systemic sclerosis, sjogren syndrome and seronegative spondyloarthropathy.
Suto T, Karonitsch T. The immunobiology of mTOR in autoimmunity. J Autoimmun. 2020 Jun;110:102373. doi: 10.1016/j.jaut.2019.102373. Epub 2019 Dec 9. PMID: 31831256.

Page 9, Line 220: It will be appropriate to write, “(2) signaling pathway MRGs,..”

Page 2, Line 55 and Page 13, Line 369: The referencing format needs to be consistent with other references. https://pubmed.ncbi.nlm.nih.gov/109750/
Suggestion: In text cite (Ohno, 1978) and in reference Ohno, S. (1978). Major sex-determining genes. Monogr Endocrinol 11, 1-140.

Table 1: A suggestion to incorporate the master regulator Group 3 innate lymphoid cells. Group 3 innate lymphoid cells (ILC3s) have emerged as master regulators of intestinal health and tissue homeostasis in mammals. (Zhou W, Sonnenberg GF. Activation and Suppression of Group 3 Innate Lymphoid Cells in the Gut. Trends Immunol. 2020 Jul 6:S1471-4906(20)30133-2. doi: 10.1016/j.it.2020.06.009. Epub ahead of print. PMID: 32646594.)

Table 2: It will be more relevant if the recent articles are referenced. For instance, NF-κB signaling pathway: Schnappauf O, Aksentijevich I. Mendelian diseases of dysregulated canonical NF-κB signaling: From immunodeficiency to inflammation. J Leukoc Biol. 2020 Jul 17. doi: 10.1002/JLB.2MR0520-166R. Epub ahead of print. PMID: 32678922.
For Yap and Hypoxia signaling pathway: Zhao C, et al. Yes-associated protein (YAP) and transcriptional coactivator with a PDZ-binding motif (TAZ): a nexus between hypoxia and cancer. Acta Pharm Sin B. 2020 Jun;10(6):947-960. doi: 10.1016/j.apsb.2019.12.010. Epub 2019 Dec 19. PMID: 32642404; PMCID: PMC7332664.

Supplementary Table 1: It will be more relevant if the recent articles are referenced. For instance, regarding MicroRNAs the following article is available. Liang Y, et al. MicroRNAs Modulate Drug Resistance-Related Mechanisms in Hepatocellular Carcinoma. Front Oncol. 2020 Jun 30;10:920. doi: 10.3389/fonc.2020.00920. PMID: 32695666; PMCID: PMC7338562.
Similarly, for NRF2 many recent articles are available. For example, Cores Á, et al. NRF2 Regulation Processes as a Source of Potential Drug Targets against Neurodegenerative Diseases. Biomolecules. 2020 Jun 14;10(6):E904. doi: 10.3390/biom10060904. PMID: 32545924.


Other minor formatting issues:

Page 2, Line 46: “…roles in major clinical diseases..” Authors can remove extra full stop.
Page 2, Line 52: “ 2. WHAT IS THE MASTER REGULATOR GENE ?”. Here, authors can bold “2” to make it consistent with their manuscript format and it will be appropriate to write “ 2. WHAT IS A MASTER REGULATOR GENE?”.
Page 3, Line 68: “… within the cells (Witczak et al., 2008)”, Please use a full stop before starting the next sentence.
Page 6, Line 155: “ 4. REGULATION OF MAJOR DISEASES BY THE MRGs”. Here, authors can bold “4” to make it consistent with their manuscript format.

---

## Round 0.2 · accepted · Accept

The authors have meticulously worked on revising the manuscript.